

# Factors influencing unmet need for contraception amongst adolescent girls and women in Cambodia

Farwa Rizvi[1], Joanne Williams[2], Steven Bowe[3] and Elizabeth Hoban[4]

[1] Faculty of Health, School of Health and Social Development, Deakin University, Burwood, Victoria, Australia
[2] Department of Health Sciences and Biostatistics, Faculty of Health, Arts & Design, Swinburne University of Technology, Hawthorne, Victoria, Australia
[3] Deakin Biostatistics unit, Faculty of Health, School of Health and Social Development, Deakin University, Burwood, Victoria, Australia
[4] Consultant at Faculty of Health, School of Health and Social Development, Deakin University, Burwood, Victoria, Australia

Corresponding author
Farwa Rizvi, rizvifa@deakin.edu.au

## ABSTRACT

**Background**. Unmet need is the gap between women's need and their practice of using contraception. Unmet need for contraception in female adolescents and women in Cambodia is a public health concern which may lead to unintended pregnancies or abortions that can contribute to maternal morbidity and mortality.

**Methods**. Bronfenbrenner's Social Ecological Model was used as a theoretical framework to analyze data from the 2014 Cambodian Demographic and Health Survey to ascertain demographic and social factors potentially associated with unmet need for contraception. Bivariate and weighted multiple logistic regression analyses with adjusted odds ratios (AOR) were conducted for 4,823 Cambodian, sexually active females aged 15–29 years.

**Results**. The percentage of unmet need for contraception was 11.7%. At the individual level of the Social Ecological Model, there was an increased likelihood of unmet need in adolescent girls 15–19 years and women 20–24 years. Unmet need was decreased in currently employed women. At the microenvironment level, there was an increased likelihood of unmet need with the husband's desire for more children and when the decision for a woman's access to healthcare was made by someone else in the household. At the macroenvironment level, unmet need was decreased in women who could access a health facility near their residence to obtain medical care. There were no urban rural differences found in the Cambodian sample population.

**Conclusion**. Unmet need for contraception in Cambodian females adolescents and women is associated with younger age, unemployment and low personal autonomy for accessing healthcare but not with education or wealth status. There is a need to implement culturally appropriate reproductive and sexual health literacy programs to increase access to modern contraception and to raise women's autonomy.

## INTRODUCTION

Unmet need for contraception is the gap between women's desire for family planning and their practice of using contraception methods (*Bradley et al., 2012*). The international conference on population and development (ICPD) as coordinated by the United Nations Population Fund (UNFPA) was held in 1994 in Cairo (*UNFPA, 1994*). At the centre of the ICPD consensus has been the call for a global commitment to sexual and reproductive health rights and access to family planning for women, gender equality, and women empowerment which are pivotal for sustainable development (*Kanem, 2019*). In 2015, the United Nations proposed the Sustainable Development Goals (SDGs) which include a set of 17 goals with 169 associated targets to be reached by 2030 (*UN, 2015*). Universal access to sexual and reproductive health (SRH), including accessibility to contraception is an important aspect of SDGs 2030 (*UN, 2015*). Sustainable Development Goal (SDG) 3 focuses on good health and well-being at all ages and the target 3.7 of SDG 3 specifically focuses on providing SRH services (*UN, 2015*).

Cambodia is a low income, agricultural country located in the South-East Asian region, bordering Thailand, Laos, and Vietnam (*DHS, CNIS & ICF, 2015*). Cambodia's turbulent political history in the past few decades has been a major reason for a slow progress towards improving the country's reproductive health parameters (*Field et al., 2011*). The birth spacing policy in Cambodia was introduced in 1995 as part of the national family planning program in the wake of the ICPD in 1994 (*UNFPA, 1994*; *WHO, 2017*). The Cambodian national population policy was introduced in 2003 (*Vong, 2008*). Despite the efforts made by the Cambodian family planning program, the current contraception prevalence rate for modern contraception methods is 39% (*DHS, CNIS & ICF, 2015*). This indicates a large gap between women's knowledge and their practice of contraception use (*WHO, 2017*). The World Health Organisation (WHO) reports that 30% of married Cambodian women of childbearing age (15–49 years) do not want to become pregnant, but they either do not use any contraception methods or use traditional methods (*WHO, 2017*).

Unmet need for contraception in females in Cambodia and other low-to-middle-income countries (LMICs) is associated with sexually transmitted infections (STIs), unintended pregnancies and unsafe abortions as a result of risky sexual behavior or unsafe sex which contribute to maternal morbidity and mortality (*Hegde, Hoban & Nevill, 2012*; *Rizvi et al., 2020*; *Sedgh, Ashford & Hussain, 2016b*). *Bradley et al. (2012)* proposed a revised algorithm to calculate unmet need for the Demographic Health Surveys (DHS) including a complex measurement of 15 survey items (*USAID, 2015*). The result was described as the percentage of sexually active women of childbearing age, and couples who would prefer to space or limit the next pregnancy but are not using any contraception methods (*Bradley et al., 2012*). The DHS are conducted globally since 1984, providing nation-wide data in the areas of population, health, and nutrition in LMICs for purposes of monitoring and evaluation programs (*USAID, 2015*).

Gender inequality is one of the challenges to sustainable development in most LMICs in East-Asia and South-East Asia, including Cambodia, Myanmar, Lao PDR, Timor-Leste,

Indonesia, Vietnam and the Philippines (*UNICEF, 2019*). The disproportionate burden of SRH issues and high unmet need in female adolescents and young people in Cambodia and other LMICs in the Asia-Pacific region is often made worse by lack of health literacy, and non-existent SRH services (*Chandra-Mouli et al., 2019*; *Rizvi et al., 2020*; *Sedgh, Ashford & Hussain, 2016a*). There is strong research evidence for integrated SRH literacy and family planning services, and increased number of schools with improved education and targeted interventions to reduce school dropouts especially for female adolescents (*Azzopardi et al., 2019*). These interventions include monetary inducements to parents and youth, free school uniforms and provisions, and employment opportunities along with achieving gender equity (*Azzopardi et al., 2019*; *Sheehan et al., 2017*; *UNFPA, 2019*; *UNICEF, 2019*). Past and current research strongly supports reducing child marriages in LMICs, including disadvantaged groups such as ethnic minorities, migrants and displaced youth (*Azzopardi et al., 2019*; *UNFPA, 2019*).

 Identification of multiple factors influencing unmet need in Cambodian females presents an opportunity to implement a holistic SRH program to improve contraception rates for reducing unintended pregnancies. This is linked to improving universal access to SRH and contraception in particular, which is an objective for the SDG 3 for good health and wellbeing (*UN, 2015*). The concept of SRH and rights proposed by *Starrs et al. (2018)* in the Lancet includes the state of complete physical, mental and social wellbeing pertaining to sexuality and the reproductive system (*Starrs et al., 2018*). All individuals have a right to make informed decisions about their reproductive bodies and be able to access the SRH services (*Starrs et al., 2018*). The definition of SRH rights encompasses women's autonomy, eradication of gender violence, respect for reproductive body functions, and the relevant services and interventions which are required to address individuals' SRH needs for their overall well-being (*Starrs et al., 2018*; *UNFPA, 2019*). Box 1 provides an overview of the policy implications.

---

**Box 1.   Policy implications.**

**What is already known**
In Cambodia in 2014, 33% of Cambodian females aged 15–24 years have been using modern contraception methods and 13% females have been using traditional methods. There is a gap between knowledge and use of contraception.

**What this study adds**
There is an increased likelihood of unmet need for contraception in adolescent girls aged 15–19 years and young women aged 20–24 years compared to 25–29-year-old women. Unmet need in Cambodian females aged 15–29 years is associated with increased parity, unemployment, and low personal autonomy for accessing healthcare and deciding about the family size.

**What insights does this study provide for informing policy decision-making**
There is a need to implement culturally appropriate policies for increased SRH literacy, education, and employment opportunities to improve gender equity. Cambodia can achieve the targets set for Sustainable Development Goal 3 as proposed

---

by the United Nations for gender equality, and improved autonomy of women by implementing relevant policy and legal frameworks for;

- SRH literacy programs
- Equal opportunity education and employment prospects to reduce gender gaps
- Linking programs for adolescents' SRH and referral between schools and health facilities
- Reducing child marriages and improving access to modern contraception
- Dissemination of SRH information via mobile-phone based text messages, social media and community-outreach programs
- Increasing awareness for short-acting modern contraceptives and focusing on condom use due to the dual protection
- Reducing barriers at the user, health systems and policy levels for promoting long-acting reversible contraceptives (LARCs) including subdermal implants for females
- Ensuring acceptable, affordable and targeted adolescent and youth friendly SRH services

**Theoretical framework**

This study is theory based and uses the modified social ecological model (SEM) by *Koren & Mawn (2010)* and *Rizvi, Williams & Hoban (2019)* as adapted from Bronfenbrenner's SEM (*Bronfenbrenner, 1979*; *Koren & Mawn, 2010*; *Rizvi, Williams & Hoban, 2019*). The SEM for unmet need includes factors operating at three levels including individual, microenvironment, and macroenvironment (*Bronfenbrenner, 1979*; *Rizvi, Williams & Hoban, 2019*). These multiple factors at different levels can influence a woman's behaviour patterns for using contraception methods. The individual level includes personal characteristics like age, sociodemographic details including area of residence (urban or rural), occupation, and education. The microenvironment level includes interpersonal and societal factors like family, friends and partners; and the macroenvironment level includes relevant policy laws and regulations, media messages for family planning and distance from the health care facilities (*Bronfenbrenner, 1979*; *Rizvi, Williams & Hoban, 2019*). Using the SEM model provides an understanding of the various factors concurrently functioning at multiple levels and their association with the unmet need, as well as the identification of prospective gaps in knowledge.

**Aim**

The primary aim was to determine the social and demographic factors influencing unmet need for contraception at the individual, microenvironment and the macroenvironment levels amongst Cambodian sexually active 15–19 years old adolescent girls and 20–29 years old women using the social ecological model.

## MATERIALS & METHODS

The 2014 Cambodian Demographic and Health Survey (CDHS) is the latest, nationally represented survey which provides countrywide data (*DHS, CNIS & ICF, 2015*). This is the first study in Cambodia to use the dataset from the 2014 CDHS. Weighted data analyses were performed to ascertain factors influencing unmet need in 4,823 Cambodian sexually active females aged 15–29 years. The revised definition from 2012 DHS was used for unmet need for contraception (*Bradley et al., 2012*). The sampling frame including the list of enumeration areas (EAs) was provided by the Cambodian National Institute of Statistics (*DHS, CNIS & ICF, 2015*). Two-stage stratified sampling and probability systematic sampling were used for participants selection (*DHS, CNIS & ICF, 2015*). All the details for the survey methodology are already mentioned in another study (*Rizvi, Williams & Hoban, 2019*). The data used in our study came from the 2014 CDHS individual questionnaire (*DHS, CNIS & ICF, 2015*).

### Ethics

The 2014 CDHS dataset is freely available with deidentified information from the website for the DHS program (https://www.dhsprogram.com/data/available-datasets.cfm). The dataset was analysed after receiving approval from MEASURE head office for the DHS program (USAID), and ethics exemption was obtained from the 'Deakin University Human Research Ethics Committee (DUHREC)', Victoria, Australia (project no 2018-157). The 2014 CDHS adhered to the legal requirements of Cambodia and received ethics permission from the Cambodian Ministry of Health and written informed consent was obtained from all participants before undertaking the survey. Additional information about 2014 CDHS can be obtained from the Cambodian Ministry of Health (*Cambodia, 2017*).

### Important definitions
#### Contraceptive prevalence rate (CPR)
Percentage of sexually active women of reproductive age group aged 15–49 years (married or in a sexual union), who are currently using, or whose partner is currently using any family planning method at a specific point in time (*Bradley et al., 2012*).

#### Unmet need for contraception as the outcome (dependent variable)
Percentage of sexually active women of reproductive age group aged 15–49 years (married or in sexual union) and couples who would prefer to space or limit the next pregnancy but are not using any contraceptive methods (*Bradley et al., 2012*). Unmet need for contraception is the sum of both the unmet need for spacing and for limiting pregnancies (*Bradley et al., 2012*; *USAID, 2015*). Women with an unmet need for spacing wish to delay the next birth for a specified time (at least two years) but they are not using any contraception (*USAID, 2015*). Women with an unmet need for limiting do not want any (more) children but they are not using any contraception (*USAID, 2015*). A set of 15 different questions from the DHS make up the complex calculation of unmet need (*USAID, 2015*).

***Identification of sexually active females from the dataset of 2014 CDHS***
Those females aged 15–29 years who gave a positive response to the question 'age at first sex' were classified as sexually active.

***Multiple independent variables***
The following independent categorical variables were identified in the literature as likely predictors, and are included in the multiple logistic regression model (*Petitet & Desclaux, 2010*; *Samandari, Speizer & O'Connell, 2010*). The variables at the individual level of SEM (*Bronfenbrenner, 1979*) included; three age groups in years (15–19, 20–24, 25–29), area of residence (rural and urban), current employment status (yes/no), parity. The variables under microenvironment level included; person in the household deciding about woman's access to healthcare and person in the household deciding about major household items purchase (woman herself, joint decision of woman and husband, husband only, someone else in the household indicating the mother-in-law/parents-in-law), husband's wish for children (both husband and wife want same number of children, husband wants more children, husband wants less children, husband does not know). Variables under the macroenvironment included; listening to any government sponsored media messages about family planning on radio (yes/no), and on television (yes/no) in the past three months, ability to access a nearby health care facility (not difficult/very difficult), participant told about family planning at the health facility (yes/no).

## Statistical analyses
The analyses included descriptive, bivariate analyses and binary logistic regression using the Stata SE version 15.1. A $P$-value $<0.05$ was considered statistically significant. The bivariate analyses included Pearson's chi square tests used for cross-tabulations to determine the degree of association between unmet need and each categorical variable. To adjust for survey cluster sampling, survey weights were applied. The odds ratios (OR) with 95% confidence interval (CI) were reported for binary logistic regression analyses showing adjusted OR (AOR). We used a forward and backward elimination approach for our model. There were missing values of 407 women in the data for two independent variables, 'person who decides for access to healthcare for the woman' and 'person who decides to purchase the major household items'. These missing values were listwise deleted, and the AOR were reported for a total of 4,416 sexually active women. There was no significant difference in the unmet need in urban or rural regions. Post-estimation diagnostic tests such as ROC curves and Hosmer-Lemeshow's goodness of fit tests were applied (*Peng, Lee & Ingersoll, 2002*).

## RESULTS
The personal, social and demographic characteristics of participants aged 15–29 years are presented in Table 1. The sample included 4,823 participants, including 1,329 (27.5%) urban and 3,494 (72.4%) rural females (15,-29 years). The sample shows 44.6% of females with primary education, 40.4% with secondary education and 10.1% with no education. The descriptive analyses show that 458 (89.4%) adolescent girls aged 15–19 years were

**Table 1** **Sociodemographic characteristics of participants aged 15–29 years.** Dataset obtained from Cambodian Demographic and Health Survey 2014 ($N = 4,823$).

| | Variables | Age group 15–19 years | Age group 20–24 years | Age group 25–29 years | Total |
|---|---|---|---|---|---|
| 1 | **Education** | | | | |
| | Higher | 3 (0.6%) | 76 (4.0%) | 151 (6.2%) | 230 (4.7%) |
| | Secondary | 242 (47.2%) | 866 (45.7%) | 842 (34.8%) | 1950 (40.4%) |
| | Primary | 225 (43.9%) | 800 (42.3%) | 1129 (46.7%) | 2154 (44.6%) |
| | No education | 42 (8.2%) | 151 (8.0%) | 296 (12.2%) | 489 (10.1%) |
| | Total | 512 (100%) | 1893 (100%) | 2418 (100%) | 4823 (100%) |
| 2 | **Wealth index** | | | | |
| | Richest | 88 (17.2%) | 455 (20.0%) | 681 (28.2%) | 1224 (25.4%) |
| | Richer | 100 (19.5%) | 355 (18.7%) | 462 (19.1%) | 917 (19.0%) |
| | Middle | 99 (19.3%) | 337 (17.8%) | 379 (15.7%) | 815 (16.9%) |
| | Poorer | 111 (21.7%) | 348 (18.4%) | 451 (18.6%) | 910 (18.8%) |
| | Poorest | 114 (22.3%) | 398 (21.0%) | 445 (18.4%) | 957 (19.8%) |
| | Total | 512 (100%) | 1893 (100%) | 2418 (100%) | 4823 (100%) |
| 3 | **Marital status** | | | | |
| | Never in union | 15 (2.9%) | 24 (1.3%) | 8 (0.3%) | 47 (0.9%) |
| | Married | 458 (89.4%) | 1722 (90.9%) | 2221 (91.8%) | 4401 (91.2%) |
| | Living with partner | 8 (1.5%) | 21 (1.1%) | 20 (0.8%) | 49 (1.0%) |
| | Widowed | 2 (0.4%) | 34 (1.8%) | 38 (1.5%) | 74 (1.5%) |
| | Divorced | 21 (4.1%) | 75 (4.0%) | 122 (5.0%) | 218 (4.5%) |
| | Separated (No longer living together) | 8 (1.5%) | 17 (0.9%) | 9 (0.3%) | 34 (0.7%) |
| | Total | 512 (100%) | 1893 (100%) | 2418 (100%) | 4823 (100%) |
| 4 | **Current employment** | | | | |
| | Yes | 304 (59.3%) | 1190 (62.8%) | 1737 (71.8%) | 3231 (67.0%) |
| | No | 208 (40.6%) | 703 (37.1%) | 680 (28.1%) | 1591 (33.0%) |
| | Total | 512 (100%) | 1893 (100%) | 2418 (100%) | 4822 (100%) |
| 5 | **Current contraception method used** | | | | |
| | Modern methods | 91 (17.7%) | 629 (33.2%) | 986 (40.7%) | 1706 (35.4%) |
| | Traditional methods | 35 (6.8%) | 245 (13%) | 378 (15.6%) | 658 (13.6%) |
| | No contraception used | 386 (75.3%) | 1019 (53.8%) | 1054 (43.5%) | 2459 (51.0%) |
| | Total | 512 (100%) | 1893 (100%) | 2418 (100%) | 4823 (100%) |
| 6 | **Parity** | | | | |
| | No children | 281 (54.9%) | 453 (23.9%) | 241 (10.0%) | 975 (20.2%) |
| | 1–2 children | 230 (44.9%) | 1385 (73.2%) | 1750 (72.4%) | 3365 (69.8%) |
| | 3 or more children | 1 (0.2) | 55 (2.9%) | 427 (17.6%) | 483 (10.0%) |
| | Total | 512 (100%) | 1893 (100%) | 2418 (100%) | 4823 (100%) |
| 7 | **Unmet need for contraception** | | | | |
| | Yes | 78 (15.2%) | 230 (12.1%) | 256 (10.6%) | 564 (11.7%) |
| | No | 434 (84.8%) | 1663 (87.9%) | 2162 (89.4%) | 4259 (88.3%) |
| | Total | 512 (100%) | 1893 (100%) | 2418 (100%) | 4823 (100%) |
| 8 | **Knowledge of ovulatory cycle** | | | | |
| | Correct information (Middle of two menstrual cycles) | 77 (15.0%) | 405 (21.4%) | 655 (27.1%) | 1137 (23.6%) |
| | Incorrect information (during periods, before periods, at any time) | 93 (18.2%) | 325 (17.2%) | 403 (16.6%) | 821 (17.0%) |
| | Do not know | 342 (66.8%) | 1162 (61.4%) | 1360 (56.3%) | 2864 (59.4%) |
| | Total | 512 (100%) | 1892 (100%) | 2418 (100%) | 4822 (100%) |

**Table 1** (*continued*)

| | Variables | Age group 15–19 years | Age group 20–24 years | Age group 25–29 years | Total |
|---|---|---|---|---|---|
| 9 | **Respondent can ask partner/husband to use a condom at sexual intercourse** (*N* = 4450) | | | | |
| | Yes | 372 (79.8%) | 1503 (86.2%) | 1936 (86.4%) | 3811 (85.6%) |
| | No | 48 (10.3%) | 122 (7.0%) | 153 (6.8%) | 323 (7.2%) |
| | Do not know | 46 (9.9%) | 118 (6.8%) | 152 (6.8%) | 316 (7.1%) |
| | Total | 466 (100%) | 1743 (100%) | 2241 (100%) | 4450 (100%) |
| 10 | **Person deciding about woman's access to healthcare** (*N* = 4448) | | | | |
| | Respondent and husband together | 244 (52.4%) | 862 (49.5%) | 1134 (50.6%) | 2240 (50.3%) |
| | Husband alone | 38 (8.2%) | 153 (8.8%) | 164 (7.3%) | 355 (8.0%) |
| | Someone else in the family | 8 (1.7%) | 18 (1.0%) | 11 (0.5%) | 37 (0.8%) |
| | Respondent alone | 175 (37.6%) | 709 (40.7%) | 932 (41.5%) | 1816 (40.8%) |
| | Total | 465 (100%) | 1742 (100%) | 2241 (100%) | 4448 (100%) |
| 11 | **Person deciding about major household items purchase** (*N* = 4446) | | | | |
| | Respondent and husband together | 354 (76.1%) | 1363 (78.3%) | 1771 (79.0%) | 3488 (78.4%) |
| | Husband alone | 32 (6.9%) | 88 (5.0%) | 103 (4.6%) | 223 (5.0%) |
| | Someone else in the family | 20 (4.3%) | 47 (2.7%) | 25 (1.1%) | 92 (2.1%) |
| | Respondent alone | 59 (12.7%) | 242 (13.9%) | 342 (15.2%) | 643 (14.4%) |
| | Total | 465 (100%) | 1740 (100%) | 2241 (100%) | 4446 (100%) |
| 12 | **Decision for family size** (*N* = 4422) | | | | |
| | Husband wants more children | 66 (14.2%) | 274 (15.8%) | 419 (18.9%) | 759 (17.1%) |
| | Husband wants fewer children | 10 (2.1%) | 93 (5.3%) | 135 (6.1%) | 238 (5.4%) |
| | Both want same number of children | 319 (68.6%) | 1162 (66.8%) | 1465 (66.0%) | 2946 (66.6%) |
| | Do not know | 70 (15.0%) | 209 (12.0%) | 200 (9.0%) | 479 (10.8%) |
| | Total | 465 (100%) | 1738 (100%) | 2219 (100%) | 4422 (100%) |
| 13 | **Participants heard about family planning media messages on radio in the last three months** | | | | |
| | Yes | 166 (32.4%) | 691 (36.5%) | 918 (38.0%) | 1775 (36.8%) |
| | No | 346 (67.6%) | 1201 (63.5%) | 1500 (62.0%) | 3047 (63.2%) |
| | Total | 512 (100%) | 1892 (100%) | 2418 (100%) | 4822 (100%) |
| 14 | **Participants heard about family planning media messages on television in the last three months** | | | | |
| | Yes | 210 (41.0%) | 905 (47.8%) | 1263 (52.2%) | 2378 (49.3%) |
| | No | 302 (59.0%) | 987 (52.2%) | 1155 (47.8%) | 2444 (50.7%) |
| | Total | 512 (100%) | 1892 (100%) | 2418 (100%) | 4822 (100%) |
| 15 | **Accessible distance to health facility and getting medical help for herself** (*N* = 4823) | | | | |
| | Not difficult | 323 (63.0%) | 1207 (63.8%) | 1630 (67.4%) | 3160 (65.5%) |
| | Very difficult | 189 (37.0%) | 686 (36.2%) | 788 (32.6%) | 1663 (34.5%) |
| | Total | 512 (100%) | 1893 (100%) | 2418 (100%) | 4823 (100%) |

**Notes.**

[a]Traditional contraception methods include; withdrawal method or coitus interruptus, abstinence, rhythm or calendar method, and other folk methods reported by the respondent including tinctures, potions, and herbs.

[b]Modern contraception methods include a) reversible methods used for short duration including oral contraceptive hormonal pills for continued monthly use, emergency contraceptive pill (morning after pill), and male and female condoms, and b) long acting reversible contraceptives (LARCs); intrauterine contraceptive devices (IUCDs), hormonal injectables, dermal implants, and c) non-reversible, permanent modern contraceptive methods including female and male sterilization. (*National Institute of Statistics/Cambodia, 2015*, and *Cahill et al 2018*).

[c]There are missing values in some variables and total number (N) is shown in the table. There are missing values of 407 women in the two 11 variables, "person to decide for respondent's health care and person deciding about major household items purchase" in the dataset.

already married at the time of survey and 67% of females aged 15–29 years were currently employed (Table 1). The results after applying binary logistic regression are shown under the social ecological model (SEM) (*Bronfenbrenner, 1979*) in a flow chart in Fig. 1.

Table 2 shows bivariate analyses using Pearson's Chi-square test used for cross-tabulations to determine the degree of association between unmet need and each categorical variable. It was noted that bivariate analyses, and Crude Odds Ratio (COR) were non-significant for the variables, 'education', and 'wealth index', so these two variables were not included in the binary logistic regression analyses. The binary logistic regression analyses results ($n = 4,416$) are presented in Table 3. The results from descriptive, bivariate and binary logistic regression analyses are presented under the individual, microenvironment and macroenvironment levels of SEM (*Bronfenbrenner, 1979*).

### Individual level of SEM

a. Knowledge of ovulation days in menstrual cycle

The participants either did not know or had incorrect information about their ovulation days during menstrual cycles (76.5%) (Table 1).

b. Contraceptive prevalence rate (CPR)

The contraceptive prevalence rate (CPR) for traditional and modern methods was 49%. Modern contraceptive prevalence rate was 35.4% and traditional contraceptive prevalence rate was 13.6%. The contraception use in the three age groups of adolescent girls aged 15–19 years, young women aged 20–24 years and 25–29-year-old women is shown in Table 1.

c. Unmet need for contraception

The unmet need was 11.7% which was the sum of unmet need for spacing (9.4%) plus unmet need for limiting (2.3%). The highest unmet need was in adolescents aged 15–19-years (15.2%), followed by women aged 20–24 years (12.1%), and 25–29 years (10.5%) (Table 1).

The results from bivariate analyses using Pearson Chi-square test in Table 2 show that *P*-value was significant ($P = 0.01$) for unmet need for the following variables; age groups, employment status, parity, husband's desire for family size, and the variable for 'person deciding for purchase of major house-hold items'.

Binary logistic regression analyses show that there was an increased likelihood of unmet need in the younger age groups including adolescent girls aged 15–19 years (AOR = 1.9, 95% CI [1.3–2.8]) and women aged 20–24 years (AOR = 1.4, 95% CI [1.1–1.8]) compared to the women in their late twenties (25–29 years) (Table 3).

d. Total number of children ever born/parity

There were 20.2% of women with no children, 69.8% of women had 1-2 children, and 10% of women had 3 or more children (Table 1). Unmet need was significantly associated with increasing parity ($P = 0.001$) (Table 2). The binary logistic regression shows increased likelihood of unmet need with multiparity (Table 3).

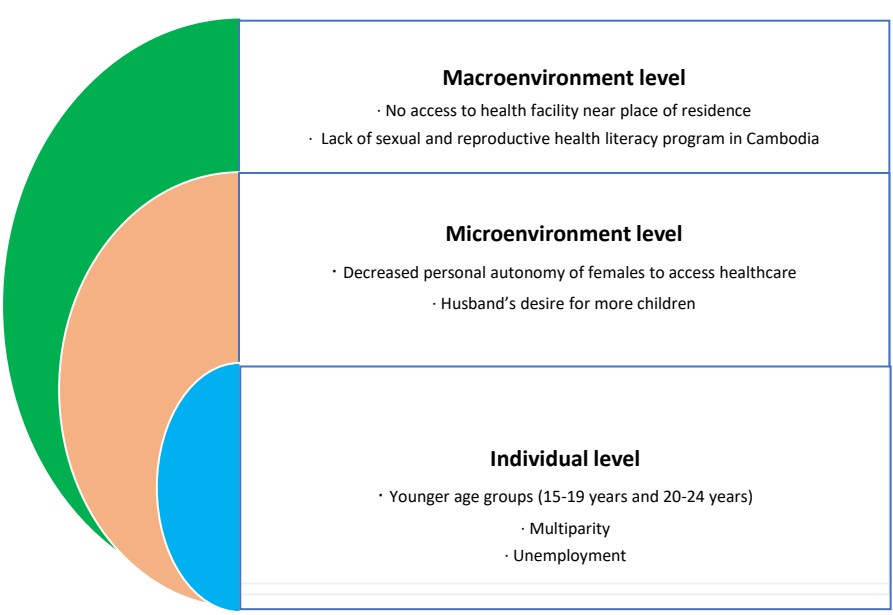

**Figure 1** Social Ecological Model for factors influencing unmet need for contraception in sexually active, single, in union, or married adolescent girls and women aged 15–29 years in Cambodia.

 e. Employment

Sixty-seven percent of women were currently employed (Table 1). Bivariate analyses show significant association ($P = 0.001$) of unmet need with employment status (Table 2). There was a decreased likelihood of unmet need in currently employed females aged 15–29 years (AOR = 0.6, 95% CI [0.5–0.8]) (Table 3).

## Microenvironment level of SEM
### *Person making decision for women's access to healthcare*
Table 1 shows the frequency distribution of females aged 15–29 years for the decision to access healthcare. There was an increased likelihood of unmet need when someone else in the household decided about the woman's access to healthcare (AOR = 2.0, 95% CI [1.1–4.1]). That person could be either the mother-in-law or the parents-in-law (Table 3).

### *Husband's desire for children*
Descriptive analyses show that 66.6% of couples wanted the same number of children, whereas 17.1% of husbands wanted more children (Table 1). Bivariate analyses show significant association ($P = 0.01$) between unmet need and decision for family size (Table 2). There was an increased likelihood of unmet need if the woman's husband wanted more children (AOR = 1.3, 95% CI [1.01–1.8]) (Table 3).

### *Woman's lack of autonomy to ask her husband to use a condom*
Descriptive analyses show that 323 (7.2%) participants reportedly could not demand condom use from their partner or husband at the time of sexual intercourse, and 316 (7.1%) were not sure if they could ask their husband to use condoms (Table 1).

**Table 2   Bivariate analyses of unmet need for contraception in Cambodian females aged 15–29 years.**

| Characteristics | No unmet need for contraception (*n*, %) | Unmet need for contraception (*n*, %) | *P*-value |
|---|---|---|---|
| **Individual level of Social Ecological Model[a]** | | | |
| **Age in years (*N = 4823*)** | | | |
| 15–19 | 434 (84.7%) | 78 (15.3%) | **0.01**[*] |
| 20–24 | 1663 (87.8%) | 230 (12.2%) | |
| 25–29 | 2162 (89.4%) | 256 (10.6%) | |
| **Region (*N = 4823*)** | | | |
| Rural | 3075 (88%) | 419 (12%) | 0.3 |
| Urban | 1184 (89%) | 145 (11%) | |
| **Wealth status (*N = 4823*)** | | | |
| Poorest | 826 (86.3%) | 131 (13.7%) | 0.2 |
| Poorer | 804 (88.3%) | 106 (11.7%) | |
| Middle | 723 (88.7%) | 92 (11.3%) | |
| Richer | 822 (89.6%) | 95 (10.4%) | |
| Richest | 1084 (88.5%) | 140 (11.5%) | |
| **Woman's current employment (*N = 4822*)** | | | |
| Yes | 2913 (90.2%) | 318 (9.8%) | **0.001**[*] |
| No | 1345 (84.5%) | 246 (15.5%) | |
| **Respondent occupation groups (*N = 4821*)** | | | |
| Not working | 928 (85.2%) | 162 (14.8%) | **0.01**[*] |
| Professional/technical | 186 (91.2%) | 18 (8.8%) | |
| Clerical | 61 (86%) | 10 (14%) | |
| Sales | 688 (90%) | 77 (10%) | |
| Agricultural/self employed | 1376 (88.3%) | 182 (11.7%) | |
| Services | 217 (88.2%) | 29 (11.8%) | |
| Skilled manual | 732 (90.1%) | 80 (9.9%) | |
| Unskilled manual | 56 (90.3%) | 6 (9.7%) | |
| Do not know | 13 (100%) | 0.0 (0%) | |
| **Women's education level (*N = 4823*)** | | | |
| No Education | 432 (88.4%) | 57 (11.6%) | 0.2 |
| Primary | 1885 (87.5%) | 269 (12.5%) | |
| Secondary | 1730 (88.7%) | 220 (11.3%) | |
| Higher | 212 (92.2%) | 18 (7.8%) | |
| **Parity (number of children) (*N = 4823*)** | | | |
| No children | 882 (90.5%) | 93 (9.5%) | **0.001**[*] |
| 1–2 children | 2975 (88.4%) | 390 (11.6%) | |
| 3 or more children | 402 (83.2%) | 81 (16.8%) | |
| **Microenvironment level of Social Ecological Model[a]** | | | |
| **Person who decides about woman's access to healthcare (*N = 4448*)** | | | |
| Respondent | 1585 (87.3%) | 231 (12.7%) | 0.1 |
| Together (husband and wife) | 1975 (88.2%) | 265 (11.8%) | |
| Husband only | 315 (88.7%) | 40 (11.3%) | |
| Someone else in the household (mother/parent in law) | 28 (75.7%) | 9 (24.3%) | |

**Table 2** (*continued*)

| Characteristics | No unmet need for contraception (*n*, %) | Unmet need for contraception (*n*, %) | *P*-value |
|---|---|---|---|
| **Person who decides about purchasing major household items** (*N* = 4446) | | | |
| Respondent | 557 (86.6%) | 86 (13.4%) | **0.01**⋆ |
| Together (husband and wife) | 3076 (88.2%) | 412 (11.8%) | |
| Husband only | 198 (88.8%) | 25 (11.2%) | |
| Someone else in the household (mother/parent in law) | 71 (77.2%) | 21 (22.8%) | |
| **Decision for family size** (*N = 4422*) | | | |
| Both want same number of children | 2610 (88.6%) | 336 (11.4%) | **0.01**⋆ |
| Husband wants more children | 649 (85.5%) | 110 (14.5%) | |
| Husband wants less children | 213 (89.5%) | 25 (10.5%) | |
| Do not know | 405 (84.5%) | 74 (15.4%) | |
| **Macroenvironment level of Social Ecological Model**[a] | | | |
| **Participants heard about family planning media messages on radio in the last three months** (*N = 4822*) | | | |
| Yes | 1575 (88.7%) | 200 (11.3%) | 0.4 |
| No | 2683 (88%) | 364 (12%) | |
| **Participants heard about family planning media messages on television in the last three months** (*N = 4822*) | | | |
| Yes | 2112 (88.8%) | 266 (11.2%) | 0.2 |
| No | 2146 (87.8%) | 298 (12.2%) | |
| **Accessible distance to health facility and getting medical help for herself** (*N = 4823*) | | | |
| Not difficult | 2805 (88.8%) | 355 (11.2%) | 0.2 |
| Very difficult | 1454 (87.4%) | 209 (12.6%) | |

**Notes.**

Pearson Chi square test was used as the statistical test of significance.

Dataset obtained from Cambodian Demographic and Health Survey 2014 (*N* = 4823), but there are some missing values in some variables in the dataset.

⋆*p*-value is significant if <0.05.

[a]Bronfenbrenners Social Ecological Model used as theoretical framework: Individual level (intrapersonal level including age, knowledge, attitudes, beliefs, practices; area of residence; employment; education and wealth status; Microenvironment level (interpersonal level including partners and peers; institutional and community level; Macroenvironment level (policy enabling, laws).

Reference: *Bronfenbrenner, 1979*.

## Macroenvironment level of SEM
### *Distance to health care facility and getting medical help*

Descriptive analyses show 1,663 (34.5%) women reported that accessing a nearby health care facility for medical help was very difficult, compared to 3,160 (65.5%) women who reported that it was not a big problem (Table 1). There was a decreased likelihood of unmet need in females aged 15–29 years who could easily access health care facility nearby to obtain medical care for themselves (AOR = 0.8, 95% CI [0.6–1.0]) (see Table 3).

## DISCUSSION

Adolescent girls and young women aged 15–24 years in Cambodia are more susceptible to having unmet need for contraception. The likelihood of unmet need is also increased in adolescent girls and women under 30 years of age with accessibility issues to a nearby

**Table 3 Binary logistic regression analyses.** Binary logistic regression analyses showing factors influencing unmet need for contraception in sexually active Cambodian females aged 15–29 years.

| Factors influencing unmet need for contraception | Adjusted Odds Ratio (AOR), 95% Confidence Interval (CI) with $p$-values ($N = 4416$) Model I |
|---|---|
| **Individual level of Social Ecological Model[a]** | |
| **Age Group** | |
| 15–19 years | 1.9 (1.3–2.8) $P = 0.001$ |
| 20–24 years | 1.4 (1.1–1.8) $P = 0.01$ |
| 25–29 years (base) | |
| **Region** | |
| Urban | 1.05 (0.7–1.5) $P = 0.7$ |
| Rural (base) | |
| **Employment** | |
| Yes | 0.6 (0.5–0.8) $P = 0.001$ |
| No (base) | |
| **Parity** | |
| 1–2 children | 1.9 (1.4–2.8) $P = 0.001$ |
| 3 or more children | 3.3 (2.1–5.3) $P = 0.001$ |
| No children (base) | |
| **Microenvironment level of Social Ecological Model[a]** | |
| **Decision for family size** | |
| Husband wants more children | 1.3 (1.0–1.8) P=0.04 |
| Husband wants fewer children | 1.1 (0.6–1.9) $P = 0.7$ |
| Do not know | 1.0 (0.7–1.5) $P = 0.9$ |
| Both want same number of children (base) | |
| **Person deciding about woman's access to healthcare ($n = 4448$ for Model I)** | |
| Respondent and husband together | 0.9 (0.7–1.2) $P = 0.6$ |
| Husband alone | 0.8 (0.5–1.3) $P = 0.5$ |
| Someone else in the family | 2.0 (1.1–4.1) $P = 0.03$ |
| Respondent alone (base) | |
| **Person deciding about major household items purchase ($n = 4446$ for Model I)** | |
| Respondent and husband together | 0.9 (0.6–1.3) $P = 0.6$ |
| Husband alone | 0.6 (0.3–1.0) $P = 0.07$ |
| Someone else in the family | 2.0 (1.0–4.1) $P = 0.05$ |
| Respondent alone (base) | |
| **Person deciding about major household items purchase ($n = 4446$ for Model I)** | |
| Respondent and husband together | 0.9 (0.6–1.3) $P = 0.6$ |
| Husband alone | 0.6 (0.3–1.0) $P = 0.07$ |
| Someone else in the family | 2.0 (1.0–4.1) $P = 0.05$ |
| Respondent alone (base) | |

health care facility and low personal autonomy when their access to healthcare is decided by someone else such as the mother-in-law, or the parents-in-law. Unmet need is increased

**Table 3** (*continued*)

| Factors influencing unmet need for contraception | Adjusted Odds Ratio (AOR), 95% Confidence Interval (CI) with *p*-values (*N* = 4416) Model I |
|---|---|
| **Macroenvironment level of Social Ecological Model**[a] | |
| **Participants heard about family planning media messages on radio in the last three months** | |
| Yes | 0.9 (0.7–1.2) *P* = 0.6 |
| No (base) | 1.0 (0.8–1.3) *P* = 0.8 |
| **Participants heard about family planning media messages on television in the last three months** | |
| Yes | |
| No (base) | |
| **Accessible distance to health facility and getting medical help for herself** (*n* = 4823 **for Model I**) | |
| Not difficult | 0.8 (0.6–1.0)  **P = 0.05** |
| Very difficult (base) | |
| **At the health facility, participants told about family planning** (*n* = 4822 **for Model I**) | |
| Yes | 1.1 (0.9–1.4) *P* = 0.3 |
| No (base) | |

**Notes.**

Model I: Number of strata = 38; Number of PSUs = 608; Number of observations = 4,416; Degree of freedom (df) = 570, $F = 4.47$, Prob > $F = 0.000$, *P*-value significant (shown in bold) if $P < 0.05$.

[a]Bronfenbrenner's social ecological model used as theoretical framework with three levels (Individual level, microenvironment level, macroenvironment level.

[b]Hosmer-Lemeshaw goodness-of-fit test for logistic model: $F(9, 562) = 0.8$, Prob > $F = 0.6$.

[c]Data used from 2014 Cambodian Demographic and Health Survey (CDHS).

[d]There are 407 missing values in the variables "person to decide for respondent's health care" and "person deciding about major household item purchase" in the dataset.

in women with low financial autonomy who are unemployed, and with low reproductive health autonomy when their husbands want more children.

## Individual level of SEM

There is an increased trend of unmet need (15.2%) in Cambodian adolescent girls aged 15–19 years. Similar findings were shown in the analyses of the 2011 Bangladesh DHS data with a higher trend of unmet need (17%) in female adolescents aged 15–19 years and youth aged 20–24 years (*Islam, Mostofa & Islam, 2016*). *Wulifan et al. (2015)* in a scoping review of unmet need in 34 quantitative and qualitative studies in low-to-middle-income countries (LMICs) reported that unmet need is increased in adolescent girls and women below the age of 34 years, especially in Zambia and Nepal (*Wulifan et al., 2015*).

Our results show increased unmet need in women with multiparity. We propose that Cambodian younger women have an increased likelihood of unmet need as they are married either in their adolescence (child brides) or early twenties and are under social pressure to have early and repeat pregnancies. Our data shows that 89.4% of adolescent girls aged 15–19 years were already married at the time of the survey. In Cambodia, one in four women are already married by age 18 years, and half of the women are married by age 20.5 years (*DHS, CNIS & ICF, 2015*). *Coll et al. (2019)* analysed data from 73 LMICs and reported that many of the child brides wish to delay the first birth, or want birth spacing, but

these adolescent girls are influenced by the society norms for early child bearing (*Coll et al., 2019*). Cambodian adolescent girls and women 15–29 years have low personal autonomy to access healthcare which can lead to an increased likelihood of having an unintended pregnancy (*Rizvi, Williams & Hoban, 2019*). Reducing child marriages, decreasing unmet need by improving adolescents' access to SRH and modern contraception can break the cycle of adolescent pregnancies and repeat, unintended pregnancies (*UNFPA, 2019*).

Women who have current, paid employment have a decreased likelihood of having unmet need. We posit that women who are currently employed could have an intrinsic motivation to use contraception to avoid an unintended pregnancy or abortion, thus ensuring their earning potential. This is an indicator of financial autonomy and could translate to an improved reproductive health autonomy in women and result in the increased use of modern contraception methods. *Wulifan et al. (2015)* and *Rizvi, Williams & Hoban (2019)* suggest that women who are currently employed may want to space or limit their future pregnancies to allow continued gainful employment, especially in urban families (*Rizvi, Williams & Hoban, 2019*; *Wulifan et al., 2015*). Conversely, *Sedgh & Hussain (2014)* show that unemployed women usually have limited or no financial autonomy, which can translate into increased gender inequality and low reproductive health autonomy (*Sedgh & Hussain, 2014*). The United Nations Children's Fund (UNICEF) reports gender inequality in employment, education or training (NEET) in adolescent girls and women compared to boys and men, in LMICs due to pre-conceived gender roles allocating unpaid domestic chores and care work to women, and paid work to the men (*UNICEF, 2019*). Studies suggest that unemployed women usually depend on their husband or partner's income and may have low decision-making ability for their SRH matters including non-use, or infrequent use of contraception (*Sedgh & Hussain, 2014*; *Wulifan et al., 2015*).

Our study did not show any significant association of formal education levels and wealth status with unmet need. One likely explanation could be the low secondary school completion rate for girls in Cambodia, as most young girls leave school after primary education (*UNFPA & CNIS 2016*). This may indicate gender inequality in attaining secondary school education in adolescent girls compared to adolescent boys (*UNFPA & CNIS, 2016*; *UNICEF, 2019*). Another reason could be an absence of a holistic SRH literacy program inculcated in Cambodian schools' curriculum. So these youth could not make an informed decision to use contraception as they did not have the SRH literacy despite attaining formal education. Some studies from LMICs show different results. *Haq, Sakib & Talukder (2017)* from Bangladesh report that higher education level and wealth status in adolescent girls and women was significantly associated with increased contraception use (*Haq, Sakib & Talukder, 2017*). In contrast, *Ngome & Odimegwu (2014)* report that in Zimbabwe, adolescents 15–19 years with a higher education level were less inclined to use contraception (*Ngome & Odimegwu, 2014*).

## Microenvironment level of SEM

Unmet need for contraception is significantly increased in women when someone else in the household decides about their access to healthcare, thus reducing their personal autonomy. We posit that health-seeking behaviours in these younger females are deeply influenced
by the ingrained societal norms which originate from a patriarchal culture. Studies from Cambodia, as well as from many LMICs in Asia, and some African countries show that women's decision to access health care, including contraception, are usually made either by their husband or by another family member (elder) in the household (*Chandra-Mouli et al., 2014*; *Gupta et al., 2015*; *Rizvi, Williams & Hoban, 2019*) . In many situations it is the mother-in-law, or the father-in-law who make these decisions (*Gupta et al., 2015*). As a result, these females have low or non-existent decision-making ability regarding contraceptive use (*Coll et al., 2019*). *Phan (2016)* used DHS data from four South-East Asian countries including the Philippines, Cambodia, Indonesia, and Timore-Leste, and reported that employment, education status, and house-hold decision making autonomy were the three factors which consistently affected women's empowerment (*Phan, 2016*). The Sustainable Development Goals (SDGs) especially SDG 3 and 5 put emphasis on human rights, women empowerment and the right of young girls and women to achieve gender equality and access to health (*WHO, 2015*).

Our results show an increased likelihood of unmet need if the husband wanted more children, indicating that woman's desire for birth spacing or for a smaller family size is not considered. We posit a lack of communication about the desired family size between the husband and wife. *Getu Melese et al. (2016)* found in a study in Ethiopia that the husband's wish for children is significantly associated with increased unmet need and subsequent unintended pregnancy (*Getu Melese et al., 2016*). The reason could be sociocultural as children are considered as wealth in the community (*Getu Melese et al., 2016*). A study in Cambodia by *Hukin (2014)* reported similar perceptions amongst most men and elders, noting that having more children is considered to increase the financial support and family networking, and it balances the burden of care for the parents (*Hukin, 2014*). Similar findings are reported by another study which noted that the likelihood of Cambodian women using effective contraception increased three times if their husbands wanted a smaller family (*Samandari, Speizer & O'Connell, 2010*). We recommend a holistic SRH program which involves the couples, and the elders in the household to improve awareness and communication about desired family size and use of effective modern contraception. Health education programs need to take into consideration a collective decision-making approach by the women, husbands and elders in the household when designing SRH and family planning campaigns (*Gupta et al., 2015*; *Samandari, Speizer & O'Connell, 2010*). Studies from Cambodia and multiple qualitative studies from LMICs in South-East Asia and Central Asia have also reported that communication amongst the husband and wife about the ideal number of children, and the husband's support for use of effective contraception can decrease the unmet need (*Samandari, Speizer & O'Connell, 2010*; *Wulifan et al., 2015*).

## Macroenvironment level of SEM

In our study, females aged 15–29 years had low unmet need if they could physically access a nearby health care facility to obtain SRH care and contraception. A proportion of women (34.5%) found it very difficult to access a health care facility close to their place of residence. Previous studies from Cambodia show that some young women due to their

migrant status reside in the low socioeconomic peri-urban areas with limited access to modern contraception and SRH services at the healthcare facilities (*Peou, 2016*; *Webber et al., 2015*). In Cambodia and most other LMICs, there is a social stigma pertaining to the adolescents' sexual behaviour and pre-marital sex (*Peou, 2016*; *Webber et al., 2015*). This manifests as reluctance on the part of the healthcare personnel in providing SRH information and services to the adolescents and youth (*Peou, 2016*; *Webber et al., 2015*). Young people are also hesitant to seek SRH care for fear of being recognised in the community by the healthcare personnel (*Peou, 2016*; *Webber et al., 2015*). There is a dearth of person-centred SRH and counselling at the healthcare facilities for the women in Cambodia. This indicates an opportunity to train the healthcare personnel at various government and private facilities by increasing their communication skills with youth for SRH education. *Azzopardi et al. (2019)* recommend providing mandatory education about SRH and the range of modern contraception methods to the male and female youth coming into the health centres as part of the adolescent and youth friendly family planning services at the global level (*Azzopardi et al., 2019*). There is a strong need to provide long-acting reversible contraceptives (LARCs) including subdermal implants at the various health centres and pharmacies in Cambodia (*Bajracharya et al., 2016*). The subdermal implants are safe and effective for 3–5 years without the need for repeat visits for resupply, but these are still underutilised in Cambodian females (*Bajracharya et al., 2016*). The multiple barriers for modern contraception use in Cambodia need to be addressed at the user, healthcare personnel, health systems, and policy levels (*Bajracharya et al., 2016*).

At the moment, there is no SRH literacy program for adolescents and youth, especially the out-of-school adolescents and young people in Cambodia (*UNFPA, 2015*). It is imperative to ensure that adolescents and young people are given a place at the table to discuss their SRH needs, make informed decisions and that they are provided with accurate SRH information (*Chandra-Mouli et al., 2019*). Cambodia has 80% of population in the rural region (*DHS, CNIS & ICF, 2015*). A local organisation, Reproductive Health Association of Cambodia (RHAC) in Cambodia sends weekly messages about SRH and contraception as part of an outreach program to a few villages in Takeo Province on remork-moto (*UNFPA, 2014*). Remork-moto is a two wheeled vehicle managed by a single driver (*UNFPA, 2014*). Many rural people, usually adolescent girls and women attend the community education sessions for SRH conducted by community based team leaders in collaboration with RHAC (*UNFPA, 2014*). Young people need to be informed about the availability and range of SRH services in the community and health centres by trained health care personnel, selected community members, outreach workers, preferably adolescents and youth themselves (*WHO, 2012*). The community awareness and information sessions about SRH and contraception could be replicated at a larger level in Cambodia in partnership with Cambodian government and multiple international and national organisations who are already working in Cambodia. Some of these organisations include World Health Organisation (WHO), United Nations Population Fund (UNFPA), Population Council, Population Services International (PSI), RHAC and Reproductive and Child Health Alliance (RACHA) (*UNFPA, 2014*; *UNFPA, 2015*; *WHO, 2017*). *Slaymaker et al. (2020)* recommend focusing on the social-structural determinants influencing the use of modern

contraception in adolescent girls and women (*Slaymaker et al., 2020*). A good infrastructure is required for enhancing research capacity, ongoing training of local stake holders and holistic development of health systems (*Beran et al., 2017*). There is a need for continued collaboration with diverse partners and stake-holders to work towards increased gender equitable environment in Cambodia.

## CONCLUSIONS

Multiple factors influence unmet need in Cambodian females, including younger age groups, unemployment, and decreased accessibility to SRH services. The social norms in Cambodian society dictate a low or non-existent personal and reproductive health autonomy in sexually active adolescent girls and women in their twenties. In the married women, this could be partly explained by a lack of communication in the couple for SRH, contraception use, and desired number of children which is influenced by the role of husband or parents-in-law. Our study results concur with the existing literature for LMICs and contribute to the gaps in literature on unmet need amongst sexually active single and married Cambodian females.

### Limitations

The study is based on cross sectional data which cannot determine causality. However, the results help us to ascertain the factors influencing unmet need in Cambodian females at different levels of the social ecological model. The study focus has been on females only, which includes sexually active single and married adolescents and women aged 15–29 years, as the negative consequences of unmet need in terms of unintended pregnancies and induced abortions are higher in these age groups. Future studies should include the perspectives of Cambodian males about SRH and contraception to better understand the factors that may play a role in unmet need.

### Recommendations
#### *Sexual and reproductive health literacy, education, and employment opportunities to improve gender equality*
There is a need for adoption of culturally sensitive, accessible and multipronged SRH literacy program and holistic family planning services for adolescents and youth, including continuous monitoring and evaluation strategies. Policies need to be implemented for decreasing the gender gaps in male and female youth, men and women for formal education, and meaningful employment opportunities in the society.

#### *Focus on awareness and availability of modern contraceptives*
Focused efforts are needed to promote increased condom use as it gives dual protection from sexually transmitted infections (STIs) and unintended pregnancies. There is a need to remove barriers at the user and supply side for the provision of modern, short-acting contraception methods including oral pills and LARCs such as subdermal implants. The subdermal implants should be made available at the pharmacies and health centres in Cambodia to the female youth and women. A voucher scheme could be started as part

of family planning program especially for making the subdermal implants financially accessible to young Cambodian women.

### Reducing child marriages

Robust policy and legal frameworks need to be implemented for reducing child marriages, adolescent pregnancies, and improving access to modern contraception for youth. These youth include both at-school and out-of-school and at-risk or displaced young females in the community. One way to reduce child marriages could be to implement policies and regulations for providing financial incentives to parents and adolescent girls to continue education till year 12. The SRH literacy could be incorporated as a mandatory aspect of the school curriculum across Cambodian schools. There is a need for a linking program for adolescents' SRH and referrals between schools and health centres in Cambodia.

### Dissemination of SRH information via social media and community-outreach programs

There is a good window of opportunity for a targeted SRH awareness program for Cambodian youth about the advantages of modern contraception. These SRH messages could be publicised via social media platform and electronic media. This could include awareness campaigns in the form of mobile-phone text messages as part of sponsored family planning program by Cambodian government. The hard-to-reach, rural youth could be sent the SRH and family planning messages via remork-moto as part of the community outreach program. Young people could be selected from the community for SRH training purposes and employed on casual basis for the outreach programs.

### Adolescent and youth friendly SRH services

Policies need to be implemented for the accessible, affordable, and acceptable contraception provision for youth as part of the SRH program. There is a need for multi-tiered implementation of targeted, evidence-based protocols and record-keeping for adolescent and youth friendly SRH services. The SRH services centres need to ensure privacy, confidentiality and swift consultancies for all young people with minimum waiting time. There could be trainers working in tandem with the healthcare personnel to ensure communication skills with young people, and channels of referral as a part of linking program between schools and health facilities, and program evaluation for improvements.

Adolescents coming to the health centres should be provided with information material for SRH and modern contraception and told about family planning methods by trained healthcare personnel. The education and information material should be displayed as posters and available as booklets and leaflets for young people. Cambodia can achieve the targets set for Sustainable Development Goal 3 as proposed by the United Nations for gender equality, and improved personal, financial and reproductive health autonomy of women by increasing social awareness amongst youth, women, and men. We recommend

robust local governance, improved leadership, ensuring participation of adolescents and youth in the community programs for SRH, and accountability at all hierarchical levels.

### Funding
The authors received no funding for this work.

### Competing Interests
The authors declare there are no competing interests.

### Author Contributions
- Farwa Rizvi conceived and designed the experiments, performed the experiments, analyzed the data, prepared figures and/or tables, authored or reviewed drafts of the paper, and approved the final draft.
- Joanne Williams conceived and designed the experiments, authored or reviewed drafts of the paper, and approved the final draft.
- Steven Bowe designed the experiments, authored or reviewed drafts of the paper, and approved the final draft.
- Elizabeth Hoban conceived and designed the experiments, authored or reviewed drafts of the paper, and approved the final draft.

### Human Ethics
The following information was supplied relating to ethical approvals (i.e., approving body and any reference numbers):

The dataset was analysed after receiving approval from MEASURE head office for the DHS program, and ethics exemption was obtained from the 'Deakin University Human Research Ethics Committee (DUHREC)', Victoria, Australia (project no 2018-157). The 2014 CDHS dataset is freely available with deidentified information from the website for the DHS program 'https://www.dhsprogram.com/data/available-datasets.cfm'.

The 2014 CDHS adhered to the legal requirements of Cambodia and received ethics permission from the Cambodian Ministry of Health and informed verbal consent was obtained from all participants before undertaking the survey.

### Data Availability
The 2014 Cambodian Demographic and Health Survey (CDHS) dataset is available with deidentified information from the Demographic and Health Survey (DHS) program.

https://www.dhsprogram.com/data/available-datasets.cfm.

USAID. The DHS Program- Demographic and Health Surveys available datasets. DOI: https://dhsprogram.com/what-we-do/survey/survey-display-464.cfm.

Additional information is available from the Cambodian National Institute of Statistics, *National Institute of Statistics/Cambodia & UNFPA, 2015*

National Institute of Statistics. Ministry of Planning, Phnom Penh, Cambodia 2017: https://translate.google.com.au/translate?hl=en&sl=km&u=http://www.nis.gov.kh/index.php/en/&prev=search.

## Supplemental Information

Supplemental information for this article can be found online at http://dx.doi.org/10.7717/peerj.10065#supplemental-information.

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
