# Peer review of "Factors influencing unmet need for contraception amongst adolescent girls and women in Cambodia"

_PeerJ, doi:10.7717/peerj.10065_

## Round 0.1 · original submission · Major Revisions

I encourage you to analyze the comments of the reviewers and submit a revision.

·

Basic reporting

Overall this is a well written article which has some valuable insights into a key determinant of health – unmet need for contraception. Below are some recommendations to improve readability.

The abstract is clear and concise.

I’m not a fan of the acronym you have used throughout: UC for unmet need. It impacts the readability of the text. In text I would strongly recommend using unmet need - particularly in intro and discussion. Perhaps use UMN if you really need to use an acronym within the results section where it would be most useful (I have seen UMN used in other manuscripts, but it is still not overly common). I realise it is very tempting with word counts being so strict, but UC is very distracting. My mind kept going to universal health care/ universal coverage. Also, in some tables you wrote UNC for unmet need which is inconsistent. I think it is fine to say: unmet need for contraception (hereafter, unmet need). That way you are not writing out the full definition each time, but it will read better.

While I agree with your statements in lines 134-137, it feels slightly clunky here. Here are two suggestions that might work:
1) Perhaps this argument fits better in the introduction paragraph about unmet need in Cambodia. Specifically, where you mention the CDHS and note young women and girls are bearing the disproportionate burden of SRH issues…but then, as you have noted, are often not included in data collection unless they are married. Hence the suitability of CDHS which has captured this information.
2) Alternatively, it could work in the methods in the context of explaining why you used the DHS data. A fair amount of space is used describing the methods used in the CDHS which are documented elsewhere (lines 118-126) – perhaps this could be written more succinctly, and the space could be used to explain why you used the CDHS data. The fact that it is nationally representative is not clear – but should be.

The description of the outcome variable (lines 163-176) is long, and really appears to repeat what is said more succinctly in lines 155-161. If you do think this much detail is required – perhaps it is better placed in a clear flow chart or table and moved to the appendix if you have met your max number of figures already.

The description of the statistical analyses is good, clear and concise. Covers the main issues with an mlr model, such as how the variables were introduced to the model, how you dealt with multicollinearity, missing data and goodness of fit.

I really like figure 1, clear and straightforward. I would switch the a, b, c to bullet points because underneath you have references labelled a, b, which could be confusing and lead people to think they are related.

Having the references mid-sentence makes the paper hard to read. Is this a PeerJ requirement? If not, I would consider moving all references to the end of the sentence, unless they are using an in sentence citation, such as “Bradley et al in 2012 proposed….”
e.g. currently: The birth spacing policy in Cambodia was introduced in 1995 as part of the national family planning program (8) in the wake of the ICPD 1994 (2).
Change to:
The birth spacing policy in Cambodia was introduced in 1995 as part of the national family planning program in the wake of the ICPD 1994 (2, 8).

Appendix - S1 policy implications - refers throughout to 'this chapter' if this is to be published with the paper it should probably be reworded to 'this paper' or this study/ analysis.

Experimental design

The research is novel and is within the scope of the journal. Research question is clear and fills an identified knowledge gap.

Ethical standard is sufficient for a secondary analysis of publicly available data. Methods are clear and sufficient for replication, with changes noted below.

Table 1 – I would like to see a total column for each age, given that age was a significant factor but urban/rural was not. It also helps with replicability; I know it can be calculated but it’s good to lay out for the reader. I appreciate you included a graph for unmet need by age in supplemental materials- thank you- but it still would be good to see those totals in the table which will be in text.

Thank you for providing the stata output. It would be more useful to readers if it had some further annotation around the variables in each model. This can be easily rectified with a small table at the beginning of the output, so the reader knows which variables relate to which question in the survey.

Validity of the findings

The data used are robust, the statistical methods are sound, and the results are transparent. The findings present a clear benefit to the literature. The conclusions and recommendations need some further refinement, as detailed in general comments below.

Additional comments

Overall this paper is well written and informative. Country level analyses like those presented here have the potential to make a significant contribution to policy direction and research priorities in LMICs. You have presented some interesting and novel findings, interpreted them well, and made some good recommendations. I believe the manuscript could be strengthened with these recommendations:

1. Is there an opportunity to bring on Cambodian collaborator/s as authors on the paper? University contacts or Non-gov institutions such as WHO, Unicef, UNFPA present good opportunities – if not for direct collaboration, then for recommendations for SRH/ adolescent health experts in Cambodia who you could collaborate with. With in-country collaborators you will be able to strengthen your recommendations/ ensure they are aligned with need and ensure better dissemination. Engagement in this way will help with enabling of translation to action. Appreciate it is not always easy to find willing and appropriate collaborators, and this does slow down the editorial process, but the opportunity for your study to be impactful is much greater. I suspect if you are doing considerable work with a focus on Cambodia you would have a few key people you would be able to contact. See reference: Beran D, Byass P, Gbakima A, et al. Research capacity building-obligations for global health partners. Lancet Glob Health 2017;5:e567-e568. doi:10.1016/S2214-109X(17)30180-8.

2. I wasn’t completely convinced by your recommendations. You had some really interesting findings, and your discussion is strong… you are well placed to make some clear and focussed recommendations for policy, intervention, and future research. I wanted to be given some clearer steps for action. Improving SRH literacy is key but how? One point that you touch on briefly is the inclusion of adol & young people in the discussion about SRH needs. This is an important and overarching issue. Especially given what you have shown in the results around autonomy of decision making in general. One way to make some clear recommendations might be to structure them around your theoretical framework. Or link back to how the recommendations can cover the issues raised across the framework. The recommendations are often the hardest part to get right – but very important for communicating your key messages. The strength of the rest of the paper makes it worth the extra work.

Some suggestions/ citations for the in pdf comments:
Employment & education are key - have a look at NEET data and Education completion data for Cambodia and neighbours (e.g. Viet Nam, Lao PDR and Thailand).
These relevant studies/ reports provide some good context and data:
Azzopardi PS, Hearps SJC, Francis KL, et al. Progress in adolescent health and wellbeing: tracking 12 headline indicators for 195 countries and territories, 1990-2016. Lancet 2019;393:1101-1118. doi:10.1016/S0140-6736(18)32427-9. (Has comparison of NEET/ Education/ demand satisfied/ child marriage for Cambodia and relevant neighbours)

Slaymaker E, Scott RH, Palmer MJ, et al. Trends in sexual activity and demand for and use of modern contraceptive methods in 74 countries: a retrospective analysis of nationally representative surveys [published correction appears in Lancet Glob Health. 2020 May;8(5):e648]. Lancet Glob Health. 2020;8(4):e567-e579. doi:10.1016/S2214-109X(20)30060-7 (recent publication which I think confirms aspects of your findings – specifically around the need to focus on those social-structural determinants).

Gender Counts, South East Asia. See: https://www.unicef.org/eap/sites/unicef.org.eap/files/2019-08/Unicef_East%20Asia_Gender%20Counts%20report_Digital_1.pdf

Discussion could be strengthened by recommendations around Adolescent friendly health services as noted in pdf – see references:
World Health Organization. Making health services adolescent friendly: Developing national quality standards for adolescent friendly health services. 2012. Available at: https://www.who.int/maternal_child_adolescent/documents/adolescent_friendly_services/en/ (this is an older but useful overview of adol friendly health services and why they are important).

Sheehan P, Sweeny K, Rasmussen B, et al. Building the foundations for sustainable development: a case for global investment in the capabilities of adolescents. Lancet 2017;390:1792-1806. doi:10.1016/S0140-6736(17)30872-3. (not specifically related to adolescent friendly health services - this is paper which shows why investment in adolescent health and wellbeing has such a good return on investment for low resource settings – with a specific focus on gender inequity. While this might be a bit further away from the focus on your paper it is a strong argument for the need to invest in SRH in adolescents).

·

Basic reporting

Let me congratulate the authors for hard working on secondary data analysis where most of such data are not utilize regularly in most of the countries.
I feel, the language needs further improvement to convey message clearly to audiences.
The introduction part contains limited information and missing information on influential factors of unmet need for family planning. I suggest to present these factors in accordance to socioecological model (SEM) and on previous similar studies in the country or your neighborhood and globally. Simply you can follow Fig 1 for a systemic literature review.
Line 113-115: it is better to have specific objectives e.g. to explore microenvironment factors of unmet need for family planning and to determine macro-environment.
Material and Methods: this section is not structured well. You can briefly describe method applied in DHS and how you select your sample for analysis. Furthermore it is important to
Line 118-139: there is much information that does not necessary to be in the method and material section.
a. The sentence begin in line 119 and end in line 122
b. Information provided in line 125-131 seems to be also irrelevant.
c. Line 132: the name of data file should be cross checked. It is better to use DHS common word that is “individual questionnaire” or in data set it is called individual record.
d. Line 132-139: information provided in these lines, if necessary, could be shifted to relevant section e.g. important definitions.
Line 155-176: information on definition for unmet need is repeated twice. It is better to present precise definition.
Line 192. The sentence about sample size already been presented in material and methods section. There is no need to repeat information.
Line 193: instead of multiple logistic regressions, it is better to use binary logistic regression.
Line 194: There is no need for reference.
Line 198: If we are presenting weighted data, there is no need to present crude rate.
Line 199-208: there is additional information on statistical tests and procedure. I propose to omit these information and just name them.
Line 155-176: information on definition for unmet need is repeated twice. It is better to present precise definition.
Line 192. The sentence about sample size already been presented in material and methods section. There is no need to repeat information.
Line 193: instead of multiple logistic regressions, it is better to use binary logistic regression.
Line 194: There is no need for reference.
Line 198: If we are presenting weighted data, there is no need to present crude rate.
Line 199-208: there is additional information on statistical tests and procedure. I propose to just name them and no need to provide such information for readers.
Line 211: Socio demographic characteristics of sample are missing. I suggest presenting such characteristics in frequency distribution table along with brief description.
Line 215: it is better to present bivariate analysis by outcome in table along with brief description. It is important to put outcome result in column while row will contain independent variables.
Line 221-224: there is no table to support these finding.
Line 225-228: However, from table 1 it is possible to calculate current use of contraceptive, but such figure is not presented.
Line 237: the AOR and its CI were not matched with table 2 result.
Line 241: No results are presented in binary logistic regression table (table 2).

Experimental design

Data are collected from a cross sectional survey. However the design is not appropriate to infer casual relationship but can predict or estimate the possible association between factors. Therefore the design is seems to be appropriate.
There were small methodological points for improvement that are just presented in the basic report.

Validity of the findings

In order of novelty, many studies conducted outside of Cambodia and as per author statement, this study is first study investigating factors connected to unmet need for family planning. Below are some points to be considered for validating and presenting finding:
Finding of this study is not presented in systematic way. It is better to start from brief sample description followed by bivariate analysis and then conduct binary logistic regression.
Table 1 is not presented in good way. I don’t think that column should have age and residence categories. This table has some typo error for example figure stands for number of urban women aged 15-19 has decimal. However, most of independent variables does not sum up to 4823 probably due to missing information but surprisingly the number of children ever born was exceed from 4823. In addition the last category of the number of children ever born should be ≥3 not > 3.
Table 2 should present only adjusted odd ratio and I think no need to present crude OR. In addition parity variable should have different categories as presented in table 1 e.g. have no children, one child, two child and three and more children. Furthermore, the symbol use for p-value should be P (capital), small p represent proportion in statistics.
Line 254-256: no figure presented as table.
Line 258-260: if there is no difference according between receiving message through media and unmet need for family planning, should not be reported.
Line 262-264: information is not translated into the table form.
Line 267-270: figures in text are not presented in table.

---

## Round 0.2 · accepted · Accept

Thank you for making the recommended revisions. Congratulations on acceptance of your manuscript.